# Endo Belly: What Is It and Why Does It Happen?—A Narrative Review

**DOI:** 10.3390/jcm12227176

**Published:** 2023-11-19

**Authors:** Renata Voltolini Velho, Franziska Werner, Sylvia Mechsner

**Affiliations:** Endometriosis Research Center Charité, Department of Gynecology Charité with Center of Oncological Surgery, Campus Virchow-Klinikum, Augustenburger Platz 1, 13353 Berlin, Germany; renata.voltolini-velho@charite.de (R.V.V.); franziska.werner@charite.de (F.W.)

**Keywords:** endometriosis, enteric nervous system, intestinal disorder, irritable bowel syndrome, microbiome, diet

## Abstract

Endometriosis is a chronic inflammatory disease where endometrial-like lesions settle outside the uterus, resulting in extensive inflammatory reactions. It is a complex disease that presents with a range of symptoms, with pain and infertility being the most common. Along with severe dysmenorrhea, cyclic and acyclic lower abdominal pain, cyclic dysuria and dyschezia, dyspareunia, and infertility, there are also nonspecific complaints that can cause confusion and make endometriosis the chameleon among gynecological diseases. These symptoms include unspecific intestinal complaints, cyclic diarrhea, but also constipation, nausea, vomiting, and stomach complaints. It appears that in addition to general bowel symptoms, there are also specific symptoms related to endometriosis such as cyclic bloating of the abdomen, known as endo belly. During the second half of the menstrual cycle leading up to menstruation, the abdomen becomes increasingly bloated causing discomfort and pain due to elevated sensitivity of the intestinal wall. Patients with endometriosis exhibit a reduced stretch pain threshold of the intestinal wall. Here, we review the endo belly, for the first time, pathophysiology and the influence of other diseases (such as irritable bowel syndrome—IBS), microbiome, hormonal levels, inflammation, and diet on the presentation of this condition.

## 1. Introduction

Endometriosis is a chronic inflammatory disease where endometrial-like lesions settle outside the uterus, resulting in extensive inflammatory reactions [1,2,3]. It is a complex condition, and its pathogenesis involves several factors that are not yet fully understood. While there have been many hypotheses regarding the development and progression of the condition, none of them provide a complete explanation for all of its manifestations. Strong evidence suggests that stem cells play a crucial role in the pathogenesis of endometriosis, but the source of these stem cells is still unknown. Genetic predisposition, along with epigenetic events, contributes to the aggressive nature of the disease [4]. Therefore, the different symptomatology between patients with endometriosis is expected to result from the different genetic, epigenetic, and molecular regulations of different endometriosis subtypes. Endometriosis presents a range of symptoms, with pain and infertility being the most common. Along with severe dysmenorrhea, cyclic and acyclic lower abdominal pain, cyclic dysuria and dyschezia, dyspareunia, and infertility, there are also nonspecific complaints that can cause confusion and make endometriosis the chameleon among gynecological diseases [5]. In rarer cases, endometriosis may also coexist with neoplastic conditions like neuroendocrine tumors [6]. These symptoms include bladder complaints; unspecific intestinal complaints, cyclic diarrhea, but also constipation, cyclic flatulence, irritable bowel symptoms; spotting, heavy bleeding; accompanying vegetative symptoms such as nausea, vomiting, stomach complaints; headache, dizziness; pain around ovulation; irregular lower abdominal pain; back pain; radiation of pain into the legs; chronic fatigue; and all types of pain [7,8]. Furthermore, endometriosis is considered a systemic disease with multisystem involvement [9,10], which is associated with a variety of comorbidities, particularly diseases associated with alterations in immune function, such as autoimmune diseases [11], allergies, or asthma [12].

It has been reported that some women may have endometriosis without experiencing any symptoms. A study by Fuentes and colleagues examined the surgical records of 287 women aged 25 to 49 who underwent surgical sterilization. Of these women, 14 (4.9%) were found to have endometriosis. Surprisingly, five out of the fourteen women had severe endometriosis, characterized by the presence of at least one endometrioma, despite being asymptomatic [13]. Another study by Shafir and team estimated that 2–43% of women who undergo tubal ligation are asymptomatic endometriosis patients [14]. It remains unclear whether these patients do not experience any symptoms or if their pain was never considered pathological by family members, partners, and physicians [7].

It appears that in addition to general bowel symptoms, there are also specific symptoms related to endometriosis such as cyclic bloating of the abdomen, known as endo belly (Figure 1). While bloating may be part of most women’s menstrual cycle, the endo belly is far more extreme. During the second half of the menstrual cycle leading up to menstruation, the abdomen becomes increasingly bloated causing discomfort and pain due to increased sensitivity of the intestinal wall. Patients with endometriosis exhibit a reduced stretch pain threshold of the intestinal wall, which is comparable to that of individuals with irritable bowel syndrome (IBS) [15].

Here, we review, for the first time, the endo belly pathophysiology and the influence of other diseases (such as IBS), microbiome, hormonal levels, inflammation, and diet on the presentation of this condition.

## 2. Endometriosis and Irritable Bowel Syndrome

Bowel complaints in endometriosis can be challenging to understand. In the past, gynecologists have primarily focused on intestinal endometriosis, which can cause mechanical issues. They would ask about stool irregularities, cyclic dyschezia, and pencil stools, but there was less emphasis on cyclic flatulence symptoms. Patients also tend not to bring up bloated abdominal problems to gynecologists because they do not seem like gynecological issues. Instead, they are more likely to see gastroenterologists who may not focus on hormone-dependent complaints. As a result, this symptomatology has been overlooked for a long time. However, in 2015, a case–cohort study found that patients with endometriosis experienced more abdominal discomfort and pain, constipation, bloating, flatulence, and imperative bowel movements than healthy women [16]. These complaints impacted the daily lives of both women with and without hormonal therapy, occurring cyclically in some and acyclically in others. The patients’ psychological well-being was significantly affected by these symptoms [16].

Endometriosis and IBS share similar symptoms and inflammation, which can lead to misdiagnosis and delayed diagnosis. Recent studies indicate that women with endometriosis are more likely to be diagnosed with IBS than women without the condition [17]. A recent meta-analysis on the prevalence of IBS in women with endometriosis included overall 96,119 subjects (11 studies), with 18,887 endometriosis patients and 77,171 controls. The odds of IBS were approximately three times higher among patients with endometriosis compared with healthy controls (odds ratio 2.97; 95% confidence interval, 2.17–4.06). Similar results were obtained after subgroup analyses by endometriosis diagnosis, IBS diagnostic criteria, and Newcastle–Ottawa Scale scores. Six studies reported prevalence rates of IBS in women with endometriosis, ranging from 10.6 to 52%. The pooled prevalence of IBS in women with endometriosis was 23.4% (95% confidence interval, 9.7–37.2) [18]. IBS can be a comorbidity with endometriosis and influences the endo belly, but it is not the cause of this endometriosis-related symptom [17].

Endo belly and the bloating experienced by individuals with IBS are similar in some ways, as both conditions can lead to abdominal discomfort and distension. However, there are key differences between the two:

(a)Underlying condition:

Endo belly is primarily associated with endometriosis; the bloating and abdominal discomfort in endo belly are often linked to the menstrual cycle and the presence of endometrial tissue in the pelvic area [19].

Bloating in IBS is associated with irritable bowel syndrome and is not directly related to the menstrual cycle; it is characterized by digestive symptoms such as abdominal pain, changes in bowel habits, and bloating [17].

(b)Triggers and patterns:

Endo belly typically worsens around the time of menstruation, as the hormonal changes during the menstrual cycle can exacerbate symptoms [20].

In IBS, bloating can occur in response to various triggers, including certain foods, stress, and gut motility issues. It may not follow a predictable monthly cycle [17,18].

(c)Diagnostic criteria:

The diagnosis of endo belly typically involves the diagnosis of endometriosis itself through medical evaluation, which may include imaging, laparoscopy, or other methods [2,7].

IBS is diagnosed based on specific criteria known as the Rome criteria, which involve the presence of recurrent abdominal pain or discomfort associated with changes in bowel habits [18]. It is important to differentiate between the two conditions because the management and treatment of endometriosis and IBS can be different [18].

## 3. Pathophysiology of Endometriosis-Associated Pain

To understand the pathophysiology of endometriosis-associated pain, it is important to comprehend how pain develops. The perception of pain involves the conversion of a biochemical signal (step 1) into a neuronal signal (step 2) through the activation of nociceptors, which sensitize sensory nerve fibers. At the spinal level, this signal is modulated (step 3) and then transmitted (attenuated/amplified) to the brain, where pain perception occurs (step 4). Steps 1 and 2 are collectively referred to as peripheral sensitization, while steps 3 and 4 are called central sensitization [1].

The focus initially is on nociceptive (inflammatory) pain, which results from the release of pain mediators from the lesions or uterine wall during menstruation or due to mechanical nerve compression or infiltration by the lesions. The pain occurs cyclically in association with menstrual bleeding and is typically experienced as abdominal pain (dysmenorrhea) [21].

Endometriosis is diagnosed on average 10 years after the onset of the first symptoms, and there is a known correlation between the delay in diagnosis and the intensity of the symptoms [22,23]. If the pain persists over a long period of time, the body’s central sensitization mechanisms are activated at the spinal level, leading to an increased release of neurotransmitters (glutamate upregulation) and expansion of the nociceptive pain field, accompanied by the lowering of the pain threshold [23]. The consequent severity of pain may go along with the occurrence of dysuria and dyschezia. As a result of endometriosis-related pain, patients search for pain-resolving positions; however, malposition leads to asymmetric reflector contraction of the pelvic floor muscle, resulting in pelvic pain and pelvic floor muscle dysfunction. In pain, patients usually tend to curve themselves and go into the fetal position [24]. These tensions can lead to secondary pelvic floor-associated pain and dyspareunia. In addition, endometriosis lesions are characterized by hyperinnervation of sensory and loss of sympathetic nerve fibers [25], which contribute to neuroinflammatory processes. In more than 60% of patients, endometriosis-associated symptoms occur before the age of 20 years [26]. The central neurological system is still very ductile in early adulthood, leading to an increasing impact of pain on brain development so regional changes have been observed in the grey matter of the brain of endometriosis patients [21].

If abdominal pain occurs during hormone therapy (with or without therapeutic amenorrhea), this may indicate increasing inflammation in the abdomen (neurogenic inflammation). Other accompanying symptoms may include nonspecific bladder and bowel dysfunction, cyclical bloating, IBS, pain radiating to the legs or back, and increasing fatigue [7]. It is not uncommon for overlapping pain syndromes to occur in the context of pain disorders, which are an expression of central sensitization [27,28]. In some cases, a somatization disorder may also be suspected.

Nonsteroidal anti-inflammatory drugs (NSAIDs) are essential in the treatment of nociceptive pain. However, in the context of more complex pain conditions, opioids are often prescribed for chronic pain conditions such as endometriosis. Treatment with opioids often leads to adverse gastrointestinal side effects such as nausea, vomiting, and dysfunction of gastrointestinal motility, known as opioid-induced bowel dysfunction. A cohort study found that the use of opioids for endometriosis-related symptoms resulted in more severe symptoms compared to the control group [16]. Ghosh et al. studied possible explanations for gastrointestinal side effects during opioid treatment and reported that the inflammatory cytokine IL-6 is involved in the morphine-induced delay of gastric emptying, which leads to acid accumulation and thus gastric inflammation [29]. However, as the exact mechanisms remain elusive, more research is needed on this topic.

Furthermore, psychological factors may have an impact on the severity and intensity of pain and the effectiveness of its treatment. Normalization of severe pain over years, misdiagnosis, and a 10-year delay in diagnosis which may lead to depression and anxiety could exacerbate the experience of pain [30].

## 4. Microbiome

The microbiome encapsulates all the genetic material of the microorganisms, including bacteria, fungi, viruses, and Archaea, that live within the host and regulate several physiological functions [31]. The influence of the microbiome on immunomodulation and the development of several inflammatory diseases is well established [32]. Much is known about how the gut microbiome maintains the integrity of the gastrointestinal epithelial lining as well as immune homeostasis, preventing bacterial translocation, which can cause low-grade systemic inflammation [33]. Immune homeostasis ensures that the immune system shows tolerance towards commensals and self-antigens but is still responsive to pathogens [34].

Contrariwise, little is known about the presence and composition of the microbiome along the female reproductive tract and its role in the development of gynecological conditions, including endometriosis [33,35]. Research has shown that endometriosis is accompanied by gut microbial dysbiosis; however, the exact findings vary among studies [35]. Bailey and Coe [36] in 2002 showed that monkeys with endometriosis had a low abundance of *Lactobacillus* and a high abundance of Gram-negative bacteria. Ata and colleagues [37] showed that patients with moderate-to-severe endometriosis (n = 14) have more *Shigella*/*Escherichia* in their colon than healthy individuals (n = 14). Contrarily, Shan et al. [38] showed that patients with the same endometriosis stage (moderate-to-severe; n = 12) had lower gut microbiota alpha diversity and a higher Firmicutes/Bacteroidetes ratio than healthy women. A recent large study on stool samples reported that the abundance of 12 bacteria in the classes *Bacilli*, *Bacteroidia*, *Clostridia*, *Coriobacteriia*, *and Gammaproteobacteria* strongly varied between endometriosis patients (n = 66) and controls (n = 198) [39]. The discrepancy between these studies shows the necessity of research that takes into account the endometriosis stage, age, race, medical history, medication use, diet, and sample studied [35]. Generally, it is believed that patients without endometriosis have high levels (>90%) of a balanced ratio of Firmicutes and Bacteroidetes and low levels of *Proteobacteria,* whereas patients with endometriosis have an imbalance of those in different intestinal segments [40].

In retrograde menstruation, shed endometrial cells and cellular debris are transported via the fallopian tubes into the abdominopelvic cavity, with 76–90% of women having retrograde menstruation [41]. Menstrual blood, as well as cell fragments in the pelvis, lead to the release of damage-associated molecular pattern molecules such as heat-shock protein 70, with iron and reactive oxygen species contributing to further activation of macrophages, mast cells, and neutrophils. These secrete proinflammatory cytokines (e.g., IL-8) and angiogenic growth factors (e.g., TNFα) into the peritoneal fluid, which support cell adhesion and thus ectopic lesion formation [40]. This process results in the development of a chronic proinflammatory environment with altered immune cell function and dysregulation, leading to a positive feedback loop and perpetuating inflammation. Dysbiosis of the gut microbiome additionally leads to inflammatory changes, increasing the permeability of the epithelial barrier [40,42]. Moreover, the gut microbiome is involved in the regulation of circulating estrogen [43], so dysbiosis may negatively affect the estrobolome and thus estrogen-dependent diseases such as endometriosis [44]. Conversely, circulating estrogen also influences dysbiosis of the gut metabolome [44], further promoting the release of proinflammatory cytokines and damage to the epithelial barrier via the gut microbiome. The complex processes between endometriosis lesions, gut dysbiosis, and inflammatory processes form a cycle and are thus maintained or further amplified (Figure 2).

These demonstrate the impact of messenger substances on the immune system and how they are affected by the microbiome. Immunological dysregulation, such as the reduced activity of natural killer cells (NK cells) in endometriosis, is a common occurrence. Uchida and colleagues conducted animal experiments that showed the administration of interleukin 12 (IL-12) improved the activity of NK cells and reduced the size of lesions. The same effect was achieved with the administration of *Lactobacillus gasseri*, which releases IL-12 [45]. In a randomized double-blind placebo-controlled study, *Lactobacillus gasseri* was found to have a positive effect on menstrual pain severity [46]. Dysbiosis can cause inflammatory activation leading to the release of lipopolysaccharides of Gram-negative bacteria, resulting in TLR-4-mediated inflammation, which may be the primary cause. Additionally, immune cell function, particularly macrophages and NK cells, is critical [42,47].

## 5. Intestinal Activity Hormonal Regulated

It is unclear why dysbiosis leads to cyclic flatulence. There must be other factors involved. Interestingly, hormonal influences affect intestinal motility and functionality. Studies show that there are significant gender-specific differences in bowel function and symptoms related to bowel diseases that affect both men and women. For instance, IBS is more common in women, with a frequency ratio of 2:1, and they experience more constipation, flatulence, abdominal pain, and incomplete emptying than men. In contrast, men with IBS report more diarrhea-associated symptoms [48,49]. Additionally, women have a longer colonic transit time than men [50,51]. Women with IBS have been found to experience cyclical variations in symptoms. During the late luteal phase, when sex hormone levels drop and are at their lowest during menstruation, there is often an exacerbation of complaints, particularly flatulence. However, during pregnancy when sex hormone levels are high, complaints are often significantly improved. In addition, during menopause, the incidence of IBS is significantly reduced, but women may suffer more from constipation [52]. Female IBS patients also report experiencing dysmenorrhea and premenstrual distress syndrome more frequently than women without this disease. These women also tend to have more gastrointestinal symptoms compared to those without dysmenorrhea [53,54,55].

It is clear that hormones have an impact on the condition. Sex hormones affect both the brain–gut connection and the physiological processes involved in IBS. They can alter how sensitive the intestines are, how they move, and their overall ability to function properly. Hormones can also impact the intestinal barrier’s strength and immune system activation in the intestinal mucosa. Moreover, gender-specific differences in stress regulation and microbiome modulation (known as microgenderome) may also play a role [52]. An Indian study by Pati et al. analyzed variations in IBS-associated complaints during the menstrual cycle and reported that more than 50% of cases were affected by flatulence in all phases of the cycle. During the luteal phase, women were more affected by constipation, while diarrhea was more common during menstruation. The majority of women experienced the feeling of incomplete defecation through all cycle phases, although the peak was reached in the follicular phase [56] (Figure 3).

## 6. Enteric Nervous System

The gut–brain axis describes the bidirectional interplay within the enteric environment between the intestinal epithelium, the mucosal immune system, and the microbiota with the enteric nervous system (ENS). This system is characterized by the presence of neurons and glia organized in ganglia arranged as two or more distinct enteric plexuses and their interconnecting neural pathways [57]. The ganglionated plexuses provide local nervous control of the tissues and cells adjacent to the ganglia, including smooth muscle, blood vessels, glands, and immune cells and tissues. The myenteric plexus (formerly, Auerbach’s plexus) lies between the longitudinal and circular muscle layers, whereas the submucosal plexus is found within the connective tissues of the submucosa. The submucosal plexus consists of two layers of ganglia, called the inner and outer submucosal plexus (also known as Meissner’s plexus and Henle’s/Schabadasch’s plexus, respectively) [57,58,59].

The ENS is comprised of different types of neurons, including motor neurons, intrinsic sensory neurons, intrinsic primary afferent neurons, and interneurons with various sensory functions. Communication between neurons is facilitated by more than 30 neurotransmitters/mediators akin to those in the central nervous system. Cholinergic and nitrergic neurotransmission is the most prominent, responsible for secretomotor activity, mucosal barrier function, and the immune response in the human intestine [60]. This is a flexible system that can adapt to changes through neuroplasticity. Various factors like age, gender, metabolism, circadian rhythm, environmental elements, trauma, inflammation, and stress can influence the ENS. Gonadotropin-releasing hormone (GnRh), estrogen, and progesterone receptors that can impact gut function as well as cyclical changes in gut motility are present in the ENS. GnRh-like peptides also appear to be released locally and act as neuromodulators in the ENS [61]. In 1994, a clinical study (phase II, randomized, double-blind, and placebo-controlled) investigated the effect of a GnRh analog, leuprolide acetate, in 30 women with symptoms of moderate to severe functional bowel disease. Lupron Depot 3.75 mg (which delivers a continuous low dose of the drug for one month) or placebo was given intramuscularly monthly for three months. Symptom scores were assessed at each four-week visit. Follicle-stimulating hormone, luteinizing hormone, estradiol, and progesterone levels were assessed before and after therapy. Patients treated with low-dose leuprolide improved progressively and significantly in scores for nausea, vomiting, bloating, abdominal pain, and early satiety and for overall symptoms. All hormone levels (except luteinizing hormone) decreased significantly [62].

It is interesting to note that innervation changes have been observed in the bowel segments of endometriosis patients who have undergone partial bowel resection for bowel endometriosis. The bowel wall shows a reduced density of sympathetic nerve fibers, indicating a neurogenic inflammation similar to what has been found in ulcerative colitis [63].

In 1998, Mathias and colleagues conducted a fascinating study that revealed the presence of a gastrointestinal motility disorder in patients with endometriosis. The study utilized antroduodenal manometry to document peristalsis and dysbiosis in the small intestine. Women with intestinal symptoms and endometriosis displayed characteristic changes in peristalsis. Out of the 50 patients examined, 40 had dysbiosis of the small intestine. The study found that a caffeine-free, low-tyramine, and low-carbohydrate diet, along with the intake of omega-3 fatty acids and clonazepam (as a GABA agonist), significantly improved the symptoms. Overall, the study suggests that gastrointestinal symptoms in patients with endometriosis stem from dysfunction in the hollow organs. Correcting the biochemical imbalance of the eicosanoid system and insulin hypersecretion with diet greatly improved the symptoms and neuromuscular dysfunction [64].

Although not part of the ENS itself, but closely connected to it, the vagus nerve is the 10th cranial nerve and the fastest and most direct route that connects the gut and the brain. It is part of the autonomic nervous system, and it plays a significant role in regulating various functions of the gastrointestinal tract, including motility, secretion, and blood flow. It serves as a major conduit for bidirectional communication between the central nervous system (CNS) and the ENS. The vagus nerve carries signals from the brain to the ENS, allowing the CNS to influence gastrointestinal functions. It also conveys sensory information from the gut to the brain, helping the body to monitor and respond to changes in the digestive system. Therefore, while the vagus nerve is not a component of the enteric nervous system itself, it is a crucial interface between the CNS and the ENS, facilitating the coordination of digestive processes and responses to various stimuli. This connection allows for the fine-tuning and regulation of gastrointestinal functions based on both external and internal cues [65]. To sum up, it is believed that patients who have endometriosis are likely impacted by their microbiome, significant inflammation in their abdomen, and changes in the ENS due to neuroplasticity. As a result of the neuroplasticity of the ENS, patients with endometriosis may experience intense intestinal symptoms due to heightened visceral sensitivity. This can be further exacerbated by hormonal factors.

## 7. Nutrition

Nutrition is increasingly recognized as a crucial factor in the disease process of endometriosis. Many patients have noticed that their nonspecific intestinal symptoms are worsened by their diet. Research on the relationship between nutrition and the risk of developing endometriosis or on the efficacy of nutritional treatments is far from complete. Nonetheless, a number of studies have raised observations that support a role for nutrition interventions, but only very few studies provide evidence-based dietary recommendations [66]. Given the complexity of the disease, this is not surprising [67]. On the internet, there are various “endometriosis diet” recommendations that may suggest a particular diet has been proven to be effective. However, some of these recommendations may not be based on experiential knowledge. Generally, a high-fiber diet rich in fruits and vegetables with reduced sugar, gluten, and animal products is recommended [68]. We have noticed that a mostly vegan diet that excludes sugar and gluten can result in a noteworthy enhancement. Additionally, there are different methods to establish and maintain a balanced gut microbiome. In an Australian study of self-management strategies in endometriosis patients, 44% of participants reported using certain dietary practices (such as gluten-free and vegan) in the last six months. The self-rated relief of pain using this method was rated 6.4 on a numerical rating scale of 0–10, which was among the most effective techniques (after cannabis and heat). In addition, 18% of patients reported being able to reduce their endometriosis-related medication by more than 50% [69].

Because of the role of inflammation in endometriosis and the fact that reducing dietary fat and increasing dietary fiber reduce circulating estrogen concentrations, the effects of plant-based diets on inflammation have been of clinical interest. In a randomized crossover trial in women with dysmenorrhea, a low-fat vegan diet was shown to increase plasma concentrations of sex hormone-binding globulin, which, in turn, would be expected to reduce estrogen activity. The diet also reduced the severity and duration of pain and moderated premenstrual symptoms [70]. Observations from the Nurses’ Health Study II, including 81,961 premenopausal women, suggested that low-glycemic index foods are associated with a reduced risk of endometriosis [71]. Increased fruit fiber has been associated with a reduced risk of endometriosis in two independent studies [72,73]. Plant-based foods contain increased amounts of polyphenols which, when metabolized into bioactive compounds, can reduce inflammation [66].

It is commonly known that inflammation can lead to the production of harmful free radicals and cause oxidative stress. Therefore, adopting an anti-inflammatory diet may prove beneficial. For instance, research has shown that consuming fish oil, which is rich in omega-3 unsaturated fatty acids, can reduce the risk of endometriosis recurrence [74]. The ratio of unsaturated to saturated fatty acids is also important in this regard [75]. Human studies that evaluate the effects of omega-3 on endometriosis patients vary between food frequency questionnaires and supplementation trials. A cohort of endometriosis patients was given various supplements including omega-3, and visual analog scale (VAS) scores were improved compared to controls [76]. Fish oil-supplemented adolescents with endometriosis were reported to have a reduction of around 50% in pain scores [75]. In a cross-sectional study of serum polyunsaturated fatty acids and clinical data from 205 women undergoing in vitro fertilization, women with higher circulating levels of eicosapentaenoic acid, one type of omega-3, were 82% less likely to have endometriosis [77].

Additionally, antioxidant vitamins C, E, and A, as well as selenium and zinc, can help to combat oxidative stress. In a recent prospective cohort study, citrus fruits, which are rich in vitamins A and C, were associated with reduced endometriosis risk [72]. Corroborating this study, Schink and collaborators found a decreased intake of vitamin B12, vitamin C, and magnesium in the endometriosis group [78]. A decrease in peripheral oxidative stress markers in women with endometriosis after vitamin C and E supplementation, which indicates the antioxidant properties of the two vitamins was also shown [79]. Apart from its anti-inflammatory properties, vitamin D is believed to have antiproliferative and immunomodulatory effects. In a case–control study, Anastasi et al. [80] reported significantly lower vitamin D serum levels associated with pelvic pain in women with endometriosis. They suggested that the effect of vitamin D could be associated with a decrease in prostaglandin synthesis, suppression of cyclo-oxygenase 2, and upregulation of 15-hydroxyl prostaglandin dehydrogenase. Furthermore, in one of the largest prospective cohort studies with 70,556 participants, Harris et al. [81] presented an inverse correlation between endometriosis and higher levels of vitamin D in plasma. The women in the highest quintile of predicted vitamin D level had a 24% lower risk of endometriosis than the women in the lowest quintile.

## 8. Strengths

This review is based on the extensive experience of Prof. Dr. Sylvia Mechsner, who has been treating endometriosis for almost 20 years. She has been heading the Endometriosis Centre Charité since 2005 and has observed that endo belly is a common complaint among patients.

## 9. Limitations

A systematic review was not reasonable as a PubMed search for “endo belly” yielded no (zero) relevant papers. A search on Google Scholar resulted in forty-five findings, out of which only nine were research papers related to psychology or that mentioned endo belly as a symptom.

Although this study offers valuable insights into endo belly and how it can be managed, it is important to acknowledge its limitations, which may affect this study’s completeness and generalizability. For instance, there is a lack of information on the number of asymptomatic patients with endometriosis, as well as a lack of clinical trials on possible medications, specific diets, and the effects of hormonal suppression, surgery, or other treatment options on endo belly. After performing laparoscopies for pain and endometriosis, we have noticed that patients experience relief for several months. Furthermore, we have observed that the endo belly often improves (temporarily) after removing all visible lesions and clearing the pelvis. Our belief is that the reduction in inflammation following surgery is linked to the improvement of the endo belly. Endo belly has never been the primary focus of previous studies and has been treated merely as one symptom of endometriosis. As a result, further research is needed to address these limitations and better inform clinical practice and patient care when it comes to endometriosis.

## 10. Conclusions

The symptomatology of endometriosis is complex and often involves complaints such as cyclic abdominal bloating, cyclic diarrhea, and nausea. It may also involve secondary diagnoses like IBS, autoimmune disorders, and depression. Since the gastrointestinal symptoms of endometriosis and IBS overlap to some extent, both conditions are characterized by inflammation, resulting in delayed diagnosis and common misdiagnosis.

However, early identification and treatment of endometriosis is crucial to prevent central sensitization mechanisms from occurring, which can lead to overlapping pain syndromes later in the course of the disease. In addition, endometriosis lesions are characterized by hyperinnervation and loss of sympathetic nerve fibers, further promoting neuroinflammatory processes. As a result, endometriosis patients exhibit alterations in the enteric nervous system, resulting in severe intestinal complaints and increased visceral sensitivity.

Moreover, dysbiosis from the intestinal microbiota contributes to inflammatory processes that in turn affect the microbiome and influence the release of further proinflammatory cytokines, as well as altered immune cell function. The resulting cycle manifests the onset and development of endometriosis. The changes in the gut microbiota of endometriosis patients also contribute to an altered estrobolome, regulating estrogen levels, further contributing to local and systemic inflammatory processes. Furthermore, sex hormones influence IBS-associated symptoms, which are subject to cyclical variations. We believe that the combination of all these processes results in the endo belly.

For improvement of intestinal symptoms, an anti-inflammatory diet, i.e., plant-based and without sugar and gluten, as well as the intake of anti-inflammatory substances is recommended. This diet may contribute both to an improvement of endometriosis-associated symptoms such as dysmenorrhea and to a reduction in endometriosis recurrences. A more specific diet for endo belly cannot be recommended due to insufficient data.

## Figures and Tables

**Figure 1 jcm-12-07176-f001:**
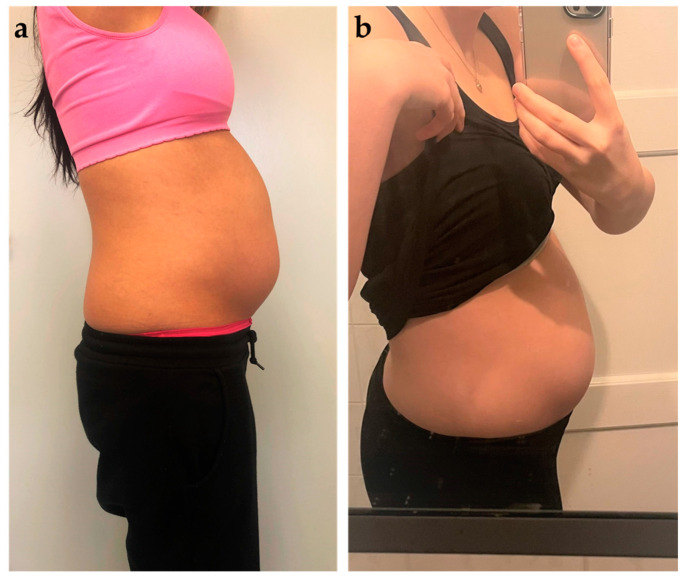
Photos (**a**,**b**) show two patients with pronounced endo belly, which usually occurs in the second half of the cycle.

**Figure 2 jcm-12-07176-f002:**
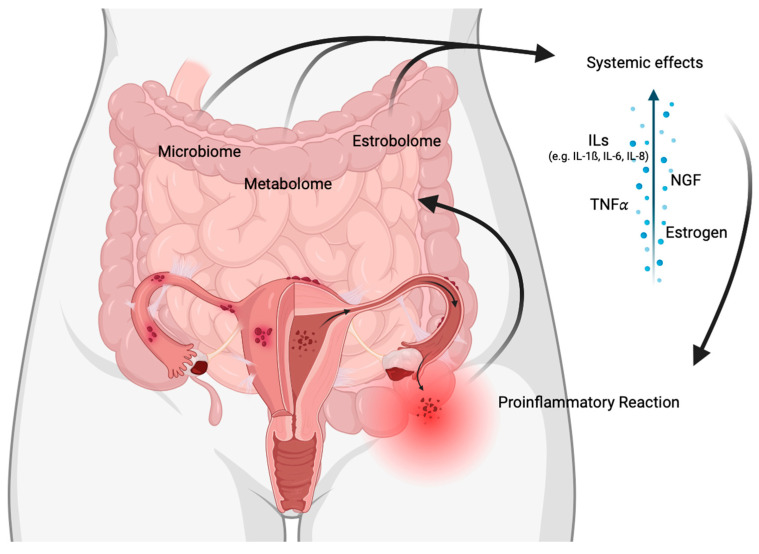
The interplay of local and systemic factors in abdominal inflammatory processes in the pathogenesis of endometriosis. The local inflammatory processes, metabolome, and estrobolome in the intestine can have systemic effects that contribute to the development of endometriosis. The pathophysiological concept underlying here is intricate and complex. Retrograde menstruation causes an increase in the number of endometrial cells and cellular debris in the pelvis, leading to a proinflammatory response that releases cytokines and growth factors. The body’s natural immune response is insufficient to break down these cells, resulting in the development of ectopic lesions. At the same time, the release of cytokines and growth factors can also disrupt gut function, causing dysbiosis. Dysbiosis may also affect intestinal epithelial cell function, stem cell activity, and the estrobolome, resulting in increased permeability that causes immunologic imbalance and low-grade systemic inflammation. This cycle can continue, with endometriosis lesions and dysbiosis exacerbating each other.

**Figure 3 jcm-12-07176-f003:**
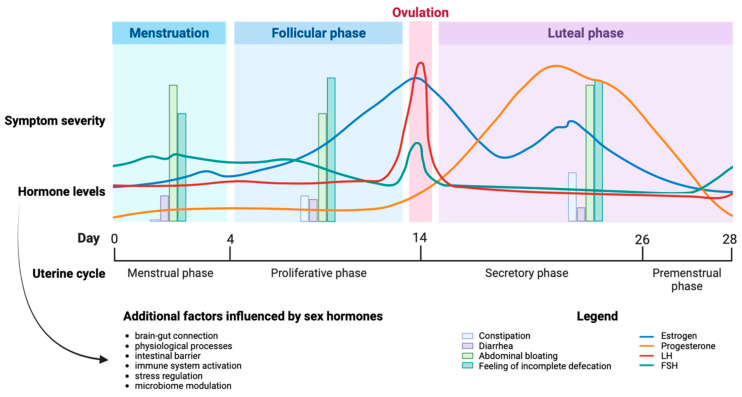
Alterations in IBS-associated complaints in the course of the menstrual cycle.

## Data Availability

Not applicable.

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
