# Peer review of "Endo Belly: What Is It and Why Does It Happen?—A Narrative Review"

_jcm, 2023, doi:10.3390/jcm12227176_

Round 1

Reviewer 1 Report

Comments and Suggestions for Authors

A well written review on a relevant topic 

 two areas that could be improved 

1) the authors give dietary recommendations, I suggest to grade the evidence of referenced papers to show the certainty these recommendations are based on. 
2) throughout the paper the reader is wondering if IBS is a co-morbidity and presents endo belly or if endo belly is an entity in itself. Please could the authors be clarify this point for each section 

Author Response

Dear Dr. Andrès,

Dear Prof. Dr. Hennerici,

as suggested by the Journal of Clinical Medicine reviewers, we are submitting the changes that have been made to our paper entitled: “Endo belly: what is it and why does it happen?” by Renata Voltolini Velho, Franziska Werner, and Sylvia Mechsner for publication as a Review.

Please find attached the answers to the reviewers. Most suggestions were accepted and included in the new version of the paper, which is attached. The modifications were highlighted in the manuscript.

                We hope to have answered all points raised by the reviewers and look forward to the publication of our manuscript in the Journal of Clinical Medicine.

We would also like to thank you for the attention that has been given to the evaluation of our paper.

Sincerely yours,

Prof. Dr. Med. Sylvia Mechsner

Reviewer 1

A well written review on a relevant topic, two areas that could be improved

1) the authors give dietary recommendations, I suggest to grade the evidence of referenced papers to show the certainty these recommendations are based on.

Answer: Thank you for your comment. A referenced paper to show the certainty of that recommendation in addition to our clinical observation was added.

2) throughout the paper, the reader wonders if IBS is a co-morbidity and presents endo belly or if it is an entity in itself. Please could the authors be clarify this point for each section

Answer: Thank you for your comment. In addition to general bowel symptoms, there are also specific symptoms related to endometriosis such as cyclic bloating of the abdomen, known as endo belly. While bloating may be part of most women's menstrual cycle, the endo belly is far more extreme. During the second half of the menstrual cycle leading up to menstruation, the abdomen becomes increasingly bloated causing discomfort and pain due to a heightened increased sensitivity of the intestinal wall. Patients with endometriosis exhibit a reduced stretch pain threshold of the intestinal wall, which is comparable to that of individuals with irritable bowel syndrome (IBS). IBS can be a comorbidity with endometriosis and influences the endo belly, but it´s not the cause of this endometriosis-related symptom. This information was added can be found in the text in lines 41-51 and 69-81.

Reviewer 2 Report

Comments and Suggestions for Authors

The authors conducted an excellent and very informative review about endo belly and its underlying etiology. I confirm that this paper is of adequate quality and of interest to the journal readers. However, I would like to comment on some issues, hoping that it may help further improving the manuscript's quality:

1) In the abstract (line 18) and the introduction (line 46), please write "elevated" or "increased" instead of "heightened sensitivity".

2) In the introduction (line 34), I would argue about spotting and heavy bleeding as symptoms of endometriosis. Those symptoms may exist in patients with endometriosis but also with coexisting adenomyosis, which is believed to be the causing factor of abnormal menstrual and uterine bleeding. Therefore, you could either delete them or specify that those symptoms are most likely to present in patients with endometriosis and adenomyosis.

3) In the introduction, I would suggest elaborating more on the pathogenesis of endometriosis. A recent article summarized the possible pathogenesis mechanisms and indicated that endometriosis is a multifactorial disease with genetic and epigenetic predisposition. Therefore, it is expected that the different symptomatology between patients with endometriosis results from the different genetic, epigenetic, and molecular regulations of different endometriosis subtypes. Please discuss and cite the following chapter: https://www.doi.org/10.1007/978-3-030-90111-0_9 

4) In the introduction (line 40), I would suggest adding that endometriosis in rarer cases may also coexist with neoplastic conditions like neuroendocrine tumors and cite the following paper: www.doi.org/10.1016/j.asjsur.2023.08.212

5) In lines 98-100, please elaborate more on the concept of "malpositioning and restraint". I could not get the idea of this part.

6) In the paragraph "Microbiome", it is always mentioned that dysbiosis affects the intestinal epithelium. However, I am wondering whether you are referring to the mucosa of the intestines or the visceral peritoneum encapsulating them. In any case, please elaborate more on the mechanism by which dysbiosis causes endo belly. Is it due to increased permeability of the visceral peritoneum or impaired absorption and increased secretion from the bowel mucosa?

7) In line 160, please write the "abdomina-pelvic cavity" or simply "the pelvic cavity" instead of "abdominal pelvis".

8) Generally speaking, is there a paper comparing the occurrence of endo belly in patients with and without bowel endometriosis? Is it more common in patients with bowel endometriosis?

Comments on the Quality of English Language

Minor language polishing is desired.

Author Response

Dear Dr. Andrès,

Dear Prof. Dr. Hennerici,

as suggested by the Journal of Clinical Medicine reviewers, we are submitting the changes that have been made to our paper entitled: “Endo belly: what is it and why does it happen?” by Renata Voltolini Velho, Franziska Werner, and Sylvia Mechsner for publication as a Review.

Please find attached the answers to the reviewers. Most suggestions were accepted and included in the new version of the paper, which is attached. The modifications were highlighted in the manuscript.

                We hope to have answered all points raised by the reviewers and look forward to the publication of our manuscript in the Journal of Clinical Medicine.

We would also like to thank you for the attention that has been given to the evaluation of our paper.

Sincerely yours,

Prof. Dr. Med. Sylvia Mechsner

Reviewer 2

The authors conducted an excellent and very informative review about endo belly and its underlying etiology. I confirm that this paper is of adequate quality and of interest to the journal readers. However, I would like to comment on some issues, hoping that it may help further improving the manuscript's quality:

1) In the abstract (line 18) and the introduction (line 46), please write "elevated" or "increased" instead of "heightened sensitivity".

Answer: Thank you for your comment. We have revised the phrasing accordingly.

2) In the introduction (line 34), I would argue about spotting and heavy bleeding as symptoms of endometriosis. Those symptoms may exist in patients with endometriosis but also with coexisting adenomyosis, which is believed to be the causing factor of abnormal menstrual and uterine bleeding. Therefore, you could either delete them or specify that those symptoms are most likely to present in patients with endometriosis and adenomyosis.

Answer: Thank you for your comment, but we believe that the text is clear when we say that spotting and heavy bleeding are part of the “nonspecific complaints that can cause confusion and make endometriosis the chameleon among gynecological diseases” (lines 30-34).

3) In the introduction, I would suggest elaborating more on the pathogenesis of endometriosis. A recent article summarized the possible pathogenesis mechanisms and indicated that endometriosis is a multifactorial disease with genetic and epigenetic predisposition. Therefore, it is expected that the different symptomatology  between  patients  with  endometriosis  results  from  the  different  genetic, epigenetic, and

molecular regulations of different endometriosis subtypes. Please discuss and cite the following chapter: https://www.doi.org/10.1007/978-3-030-90111-0_9

Answer: Thank you for your comment. Some lines about the pathogenesis mechanisms of endometriosis were added to the text (27-36).

4) In the introduction (line 40), I would suggest adding that endometriosis in rarer cases may also coexist with neoplastic conditions like neuroendocrine tumors and cite the following paper: www.doi.org/10.1016/j.asjsur.2023.08.212

Answer: Thank you for your comment. This information was added to the text (lines 40-41).

5) In lines 98-100, please elaborate more on the concept of "malpositioning and restraint". I could not get the idea of this part.

Answer: Thanks for your comment. As a result of endometriosis-related pain, patients search for pain-resolving positions, however, malposition leads to asymmetric reflector contraction of the pelvic floor muscle, resulting in pelvic pain and pelvic floor muscle dysfunction. In pain, patients usually tend to curve themselves and go to the fetal position. These are malposition. This concept is more elaborate now in lines 110-114.

6) In the paragraph "Microbiome", it is always mentioned that dysbiosis affects the intestinal epithelium. However, I am wondering whether you are referring to the mucosa of the intestines or the visceral peritoneum encapsulating them. In any case, please elaborate more on the mechanism by which dysbiosis causes endo belly. Is it due to increased permeability of the visceral peritoneum or impaired absorption and increased secretion from the bowel mucosa?

Answer: Thanks for your comment. We are referring to the mucosa of the intestines. In healthy situations, the composition of the gut microbiome is associated with a higher mucus layer thickness, the production of antimicrobial signals, and different short-chain fatty acids such as butyrate and propionate. Both butyrate and propionate bind to G protein-coupled receptors (GPR)-43 and GPR-41 expressed on the enteroendocrine L-cells thereby stimulating the secretion of gut peptides such as glucagon-like peptide-1 (GLP-1) and peptide YY (PYY). This effect contributes to reducing food intake and improving glucose metabolism. Propionate can also bind to GPR-43 expressed on lymphocytes in order to maintain appropriate immune defense. Butyrate activates peroxisome proliferator-activated receptor-γ (PPAR-γ) leading to beta-oxidation and oxygen consumption, a phenomenon contributing to maintaining anaerobic conditions in the gut lumen. Changes in the gut microbiome are linked with lower mucus thickness, decreased antimicrobial defense, and butyrate and propionate production. As a consequence, L-cells secrete fewer gut peptides. The lack of PPAR-γ activation leads to higher oxygen available for the microbiota at the proximity of the mucosa and increases the proliferation of Enterobacteriaceae. The decrease in propionate also contributes to the lower abundance of specific T cells in the lamina propria of the gut. Altogether, such changes in the microbial environment and metabolites induce a leakage of pathogen-associated molecular patterns (PAMPs) such as the lipopolysaccharide (LPS) that are increased in the blood and trigger low-grade inflammation (PMID: 29934437).

We believe that Endo Belly is caused by changes within the enteric nervous system, leading to severe intestinal issues and heightened visceral sensitivity. These changes may be a result of dysbiosis from the intestinal microbiota, which in turn contributes to inflammatory processes and influences the release of additional pro-

inflammatory cytokines. Furthermore, the altered estrobolome may play a role in regulating estrogen levels and contributing to both local and systemic inflammation. Finally, sex hormones are subject to cyclical variations which may also impact the development of Endo Belly. This summary of our idea is in the conclusion section.

7) In line 160, please write the "abdomina-pelvic cavity" or simply "the pelvic cavity" instead of "abdominal pelvis".

Answer: Thank you for your comment. We have revised the phrasing accordingly.

8) Generally speaking, is there a paper comparing the occurrence of endo belly in patients with and without bowel endometriosis? Is it more common in patients with bowel endometriosis?

Answer: Until now, endo belly was merely considered to be one of the many symptoms of endometriosis, from our knowledge, this is the first paper to fully focus on it. This means that there is no paper comparing the occurrence of endo belly in patients with and without bowel endometriosis.

Reviewer 3 Report

Comments and Suggestions for Authors

A very interesting article but in need of improvement.

Author Response

Dear Dr. Andrès,

Dear Prof. Dr. Hennerici,

as suggested by the Journal of Clinical Medicine reviewers, we are submitting the changes that have been made to our paper entitled: “Endo belly: what is it and why does it happen?” by Renata Voltolini Velho, Franziska Werner, and Sylvia Mechsner for publication as a Review.

Please find attached the answers to the reviewers. Most suggestions were accepted and included in the new version of the paper, which is attached. The modifications were highlighted in the manuscript.

                We hope to have answered all points raised by the reviewers and look forward to the publication of our manuscript in the Journal of Clinical Medicine.

We would also like to thank you for the attention that has been given to the evaluation of our paper.

Sincerely yours,

Prof. Dr. Med. Sylvia Mechsner

Reviewer 1

A well written review on a relevant topic, two areas that could be improved

1) the authors give dietary recommendations, I suggest to grade the evidence of referenced papers to show the certainty these recommendations are based on.

Answer: Thank you for your comment. A referenced paper to show the certainty of that recommendation in addition to our clinical observation was added.

2) throughout the paper, the reader wonders if IBS is a co-morbidity and presents endo belly or if it is an entity in itself. Please could the authors be clarify this point for each section

Answer: Thank you for your comment. In addition to general bowel symptoms, there are also specific symptoms related to endometriosis such as cyclic bloating of the abdomen, known as endo belly. While bloating may be part of most women's menstrual cycle, the endo belly is far more extreme. During the second half of the menstrual cycle leading up to menstruation, the abdomen becomes increasingly bloated causing discomfort and pain due to a heightened increased sensitivity of the intestinal wall. Patients with endometriosis exhibit a reduced stretch pain threshold of the intestinal wall, which is comparable to that of individuals with irritable bowel syndrome (IBS). IBS can be a comorbidity with endometriosis and influences the endo belly, but it´s not the cause of this endometriosis-related symptom. This information was added can be found in the text in lines 41-51 and 69-81.

Reviewer 2

The authors conducted an excellent and very informative review about endo belly and its underlying etiology. I confirm that this paper is of adequate quality and of interest to the journal readers. However, I would like to comment on some issues, hoping that it may help further improving the manuscript's quality:

1) In the abstract (line 18) and the introduction (line 46), please write "elevated" or "increased" instead of "heightened sensitivity".

Answer: Thank you for your comment. We have revised the phrasing accordingly.

2) In the introduction (line 34), I would argue about spotting and heavy bleeding as symptoms of endometriosis. Those symptoms may exist in patients with endometriosis but also with coexisting adenomyosis, which is believed to be the causing factor of abnormal menstrual and uterine bleeding. Therefore, you could either delete them or specify that those symptoms are most likely to present in patients with endometriosis and adenomyosis.

Answer: Thank you for your comment, but we believe that the text is clear when we say that spotting and heavy bleeding are part of the “nonspecific complaints that can cause confusion and make endometriosis the chameleon among gynecological diseases” (lines 30-34).

3) In the introduction, I would suggest elaborating more on the pathogenesis of endometriosis. A recent article summarized the possible pathogenesis mechanisms and indicated that endometriosis is a multifactorial disease with genetic and epigenetic predisposition. Therefore, it is expected that the different symptomatology  between  patients  with  endometriosis  results  from  the  different  genetic, epigenetic, and

molecular regulations of different endometriosis subtypes. Please discuss and cite the following chapter: https://www.doi.org/10.1007/978-3-030-90111-0_9

Answer: Thank you for your comment. Some lines about the pathogenesis mechanisms of endometriosis were added to the text (27-36).

4) In the introduction (line 40), I would suggest adding that endometriosis in rarer cases may also coexist with neoplastic conditions like neuroendocrine tumors and cite the following paper: www.doi.org/10.1016/j.asjsur.2023.08.212

Answer: Thank you for your comment. This information was added to the text (lines 40-41).

5) In lines 98-100, please elaborate more on the concept of "malpositioning and restraint". I could not get the idea of this part.

Answer: Thanks for your comment. As a result of endometriosis-related pain, patients search for pain-resolving positions, however, malposition leads to asymmetric reflector contraction of the pelvic floor muscle, resulting in pelvic pain and pelvic floor muscle dysfunction. In pain, patients usually tend to curve themselves and go to the fetal position. These are malposition. This concept is more elaborate now in lines 110-114.

6) In the paragraph "Microbiome", it is always mentioned that dysbiosis affects the intestinal epithelium. However, I am wondering whether you are referring to the mucosa of the intestines or the visceral peritoneum encapsulating them. In any case, please elaborate more on the mechanism by which dysbiosis causes endo belly. Is it due to increased permeability of the visceral peritoneum or impaired absorption and increased secretion from the bowel mucosa?

Answer: Thanks for your comment. We are referring to the mucosa of the intestines. In healthy situations, the composition of the gut microbiome is associated with a higher mucus layer thickness, the production of antimicrobial signals, and different short-chain fatty acids such as butyrate and propionate. Both butyrate and propionate bind to G protein-coupled receptors (GPR)-43 and GPR-41 expressed on the enteroendocrine L-cells thereby stimulating the secretion of gut peptides such as glucagon-like peptide-1 (GLP-1) and peptide YY (PYY). This effect contributes to reducing food intake and improving glucose metabolism. Propionate can also bind to GPR-43 expressed on lymphocytes in order to maintain appropriate immune defense. Butyrate activates peroxisome proliferator-activated receptor-γ (PPAR-γ) leading to beta-oxidation and oxygen consumption, a phenomenon contributing to maintaining anaerobic conditions in the gut lumen. Changes in the gut microbiome are linked with lower mucus thickness, decreased antimicrobial defense, and butyrate and propionate production. As a consequence, L-cells secrete fewer gut peptides. The lack of PPAR-γ activation leads to higher oxygen available for the microbiota at the proximity of the mucosa and increases the proliferation of Enterobacteriaceae. The decrease in propionate also contributes to the lower abundance of specific T cells in the lamina propria of the gut. Altogether, such changes in the microbial environment and metabolites induce a leakage of pathogen-associated molecular patterns (PAMPs) such as the lipopolysaccharide (LPS) that are increased in the blood and trigger low-grade inflammation (PMID: 29934437).

We believe that Endo Belly is caused by changes within the enteric nervous system, leading to severe intestinal issues and heightened visceral sensitivity. These changes may be a result of dysbiosis from the intestinal microbiota, which in turn contributes to inflammatory processes and influences the release of additional pro-

inflammatory cytokines. Furthermore, the altered estrobolome may play a role in regulating estrogen levels and contributing to both local and systemic inflammation. Finally, sex hormones are subject to cyclical variations which may also impact the development of Endo Belly. This summary of our idea is in the conclusion section.

7) In line 160, please write the "abdomina-pelvic cavity" or simply "the pelvic cavity" instead of "abdominal pelvis".

Answer: Thank you for your comment. We have revised the phrasing accordingly.

8) Generally speaking, is there a paper comparing the occurrence of endo belly in patients with and without bowel endometriosis? Is it more common in patients with bowel endometriosis?

Answer: Until now, endo belly was merely considered to be one of the many symptoms of endometriosis, from our knowledge, this is the first paper to fully focus on it. This means that there is no paper comparing the occurrence of endo belly in patients with and without bowel endometriosis.

Reviewer 3

A very interesting article but in need of improvement.

Answer: Thanks for your comment. As no suggestions were made by Reviewer 3, but by Reviewers 1 and 2, those suggestions were taken into consideration for the paper's improvement.

Round 2

Reviewer 1 Report

Comments and Suggestions for Authors

My main comment lat time was regarding the the distctinction between endo belly and IBS. The authors have added a line 88/89 stating IBS can be a co-morbidity..., but no references are added. Please add references. 

I suggest to add a short summary of what endo belly actually constitutes and how it is caused with particular attention to where it differs from IBS

Author Response

As suggested by the Journal of Clinical Medicine reviewers, we are submitting the changes that have been made to our paper entitled: “Endo belly: what is it and why does it happen?” by Renata Voltolini Velho, Franziska Werner, and Sylvia Mechsner for publication as a Review.

Please find attached the answers to the reviewers. Most suggestions were accepted and included in the new version of the paper, which is attached. The modifications were highlighted in the manuscript.

                We hope to have answered all points raised by the reviewers and look forward to the publication of our manuscript in the Journal of Clinical Medicine.

We would also like to thank you for the attention that has been given to the evaluation of our paper.

Sincerely yours,

Prof. Dr. Med. Sylvia Mechsner

Reviewer 1

My main comment lat time was regarding the the distctinction between endo belly and IBS. The authors have added a line 88/89 stating IBS can be a co-morbidity..., but no references are added. Please add references.

I suggest to add a short summary of what endo belly actually constitutes and how it is caused with particular attention to where it differs from IBS.

Answer: Thank you for your comment. A reference was added for the lines 88-90. IBS can be a comorbidity with endometriosis and influences the endo belly, but it´s not the cause of this endometriosis-related symptom [15]. A summary of what is endo belly and how it´s distinct from IBS was added in topic 2 (lines 91 – 112).

Endo belly and the bloating experienced by individuals with IBS are similar in some ways, as both conditions can lead to abdominal discomfort and distension. However, there are key differences between the two:

  1. a) Underlying condition:

Endo belly is primarily associated with endometriosis, the bloating and abdominal discomfort in endo belly are often linked to the menstrual cycle and the presence of endometrial tissue in the pelvic area [16].

Bloating in IBS is associated with irritable bowel syndrome, and is not directly related to the menstrual cycle; it is characterized by digestive symptoms such as abdominal pain, changes in bowel habits, and bloating [15].

  1. b) Triggers and patterns:

Endo belly typically worsens around the time of menstruation, as the hormonal changes during the menstrual cycle can exacerbate symptoms [17].

In IBS, bloating can occur in response to various triggers, including certain foods, stress, and gut motility issues. It may not follow a predictable monthly cycle [14, 15].

  1. c) Diagnostic criteria:

Diagnosis of endo belly typically involves the diagnosis of endometriosis itself through medical evaluation, which may include imaging, laparoscopy, or other methods [2, 7].

IBS is diagnosed based on specific criteria known as the Rome Criteria, which involve the presence of recurrent abdominal pain or discomfort associated with changes in bowel habits [15]. It's important to differentiate between the two conditions because the management and treatment of endometriosis and IBS can be different.

Reviewer 3 Report

Comments and Suggestions for Authors

The authors have not complied with any my review. I have not received a response from the authors. Was the article registered with PROSPERO?

Author Response

Reviewer 3

The authors have not complied with any my review. I have not received a response from the authors. Was the article registered with PROSPERO?

Answer: Apologies for the inconvenience, but no comments were found in the MDPI system. All comments have now been addressed.

Thank you for the opportunity to review an article that describes a very important topic.

  1. Introduction - please indicate clearly what is the aim study of this article? Please underline strongly the "study aim".

Answer: Thanks for your comment. The aim of this article is to comprehensively examine the pathophysiology of "endo belly," a condition associated with endometriosis, and to assess how various factors, including co-existing conditions like Irritable Bowel Syndrome (IBS), the microbiome, hormonal levels, inflammation, and dietary factors, influence its manifestation. This study aims to provide a detailed understanding of the factors contributing to "endo belly" and how they interconnect, shedding light on potential avenues for better management and treatment of this condition (lines 19-21 and 58-60).

  1. The authors presented selected chapter-points, but how did they review this article? Which databases were reviewed. Was the article submitted to PROSPERO? Please include the chapter "Material and methods"

Answer: Thank you for your feedback. The paper is based on the extensive experience of Prof. Dr. Sylvia Mechsner, who has been treating endometriosis for almost 20 years. She has been heading the Endometriosis Centre Charité since 2005 and has observed that endo belly is a common complaint among patients.  Unfortunately, it is not possible to conduct a systematic review on this topic as a search on Pubmed, for example, did not yield any (zero) relevant papers.

  1. "3. Pathophysiology of Endometriosis-Associated Pain" - pain in endometriosis is not just pelvic or abdominal pain. Patients also report sensations of pain in other parts of the body. It is worth adding a reference at this point: https://doi.org/10.3390/jcm12082862

Answer: Thanks for your comment, however, we added this reference in point 1, line 46.

  1. "6. Enteric nervous system" - It is worth adding information about the vagus nerve, which also innervates the digestive tract.

Answer: Thanks for your suggestion. Although not part of the ENS itself, but closely connected to it, the vagus nerve is the 10th cranial nerve and the fastest and most direct route that connects the gut and the brain. It is part of the autonomic nervous system, and it plays a significant role in regulating various functions of the gastrointestinal tract, including motility, secretion, and blood flow. It serves as a major conduit for bidirectional communication between the central nervous system (CNS) and the ENS. The vagus nerve carries signals from the brain to the ENS, allowing the CNS to influence gastrointestinal functions. It also conveys sensory information from the gut to the brain, helping the body monitor and respond to changes in the digestive system. So, while the vagus nerve is not a component of the enteric nervous system itself, it is a crucial interface between the CNS and the ENS, facilitating the coordination of digestive processes and responses to various stimuli. This connection allows for the fine-tuning and regulation of gastrointestinal functions based on both external and internal cues (63). This info was added in the text, lines 323-335.

  1. "8. Conclusion" - is not readable by the reader, please remember to use paragraphs or provide a conclusion in points.

Answer: Thank you for your suggestion. We use paragraphs to improve the clarity and organization of our writing, making it easier to understand.

  1. Missing "Limitations" please add what are the limitations of this review.

Answer: Thanks for your comment. A limitation of this review is that the paper is based on the extensive experience of Prof. Dr. Sylvia Mechsner, who has been treating endometriosis for almost 20 years. She has been heading the Endometriosis Centre Charité since 2005 and has observed that endo belly is a common complaint among patients. Unfortunately, it is not possible to conduct a systematic review on this topic as a search on Pubmed, for example, did not yield any (zero) relevant papers. This info was added in the conclusion, lines 429 – 434.

Round 3

Reviewer 3 Report

Comments and Suggestions for Authors

Thank you for the opportunity to review again. The authors have made corrections. The authors have not registered this work in PROPSERO - a pity, so please be sure to add in the title of the narrative review or or please add in the title - practical implications. I don't dare doubt that the Prof. Dr. Sylvia Mechsnerhas a wealth of knowledge and experience but the article is a review so the authors must fit within the guidelines of the review. I believe that, as it stands, the article provides a great deal of knowledge for readers

Author Response

Dear Review 3,

thanks for your comment. The title was changed.

Best,